# Cross-Talk and Subset Control of Microglia and Associated Myeloid Cells in Neurological Disorders

**DOI:** 10.3390/cells11213364

**Published:** 2022-10-25

**Authors:** Jatia Mills, Liliana Ladner, Eman Soliman, John Leonard, Paul D. Morton, Michelle H. Theus

**Affiliations:** 1Department of Biomedical Sciences and Pathobiology, Virginia Tech, Blacksburg, VA 24061, USA; 2Virginia Tech Carilion School of Medicine, Roanoke, VA 24016, USA; 3Center for Engineered Health, Virginia Tech, Blacksburg, VA 24061, USA

**Keywords:** single-cell sequencing, circulating monocytes, macrophages, neuroinflammation, neurological disease, bone marrow transplant, irradiation, neuroimaging

## Abstract

Neurological disorders are highly prevalent and often lead to chronic debilitating disease. Neuroinflammation is a major driver across the spectrum of disorders, and microglia are key mediators of this response, gaining wide acceptance as a druggable cell target. Moreover, clinical providers have limited ability to objectively quantify patient-specific changes in microglia status, which can be a predictor of illness and recovery. This necessitates the development of diagnostic biomarkers and imaging techniques to monitor microglia-mediated neuroinflammation in coordination with neurological outcomes. New insights into the polarization status of microglia have shed light on the regulation of disease progression and helped identify a modifiable target for therapeutics. Thus, the detection and monitoring of microglia activation through the inclusion of diagnostic biomarkers and imaging techniques will provide clinical tools to aid our understanding of the neurologic sequelae and improve long-term clinical care for patients. Recent achievements demonstrated by pre-clinical studies, using novel depletion and cell-targeted approaches as well as single-cell RNAseq, underscore the mechanistic players that coordinate microglial activation status and offer a future avenue for therapeutic intervention.

## 1. Distinct Role for Microglia and Other Myeloid Cells in the Neurological Disorders


 **a.** 
**Overview of the roles of CNS myeloid cells in the pathogenesis of neurological diseases**



Myeloid cells of the central nervous system (CNS) have distinct ontogeny and function in health and disease [1]. Microglia, the parenchymal mononuclear phagocytic cells of the CNS, are detected in the brain during the early stages of embryonic development following emigration from the yolk sac [2] and remain self-maintained throughout adulthood [3,4,5,6]. While microglia are the most abundant myeloid cells within the brain, an immunologically diverse population of non-parenchymal macrophages occupy the border regions of the immune-privileged CNS [7,8,9]. In response to neuroinflammation, circulating mononuclear myeloid cells, such as monocytes and dendritic cells, can infiltrate the brain and influence recovery. Unlike microglia, circulating myeloid cells are routinely replaced by hematopoietic stem cells from the bone marrow (BM) throughout life [10,11,12].

### 1.1. CNS Resident Microglia

Following birth, microglia play numerous roles essential to healthy brain development and maintenance, such as buffering of excessive and apoptotic cells, pruning of inefficient synapses, and regulating neuronal activity at the synaptic level, ultimately fostering circuit plasticity [13]. Microglia primarily accomplish this via phagocytosis, a process through which cells recognize, engulf, and digest degraded cellular particles and debris. Following acquired brain injury, microglia are capable of acting as antigen-presenting cells to initiate activation of neighboring T-cells and orchestrate an inflammatory response that can impair proper neurological function [14,15,16,17,18,19]. In addition, microglia are activated following brain injury and undergo significant morphological changes to execute different tasks. Within the healthy brain, a predominant trait of homeostatic microglia is a ramified morphology which enables continuous monitoring of local environments and fine-tuning of neuronal circuits. De-ramification of microglia occurs upon activation; microglia retract their processes to acquire a bushy-like amoeboid shape.

The contribution of microglia to the neuroinflammatory response is context-dependent and temporally regulated [2,5,12,14,15,16,18,19,20,21,22,23,24,25,26,27,28,29,30,31,32,33,34,35,36,37,38]. In the neuroinflammatory milieu, different mediators activate microglia polarization to pro-inflammatory (e.g., TNFα, INFϒ producing) or anti-inflammatory (e.g., IL-4, IL-13, IL-10) phenotypes [39]. Pro-inflammatory microglia secrete abundant amounts of inflammatory cytokines and chemokines (e.g., TNFα, IL-6, IL-1β) that not only initiate inflammation in the brain but aid in the recruitment of peripheral-derived immune cells. Together with peripheral-derived macrophages, microglia contribute to inflammation-mediated cytotoxicity of neurons, glia, and endothelium [2,15,17,21,33,40,41,42,43].

### 1.2. Peripheral-Derived Monocytes/Macrophages

The inflammatory response initiated by CNS glia is responsible for the recruitment of peripheral-derived myeloid cells, including monocytes [16,44,45,46]. Monocyte subset polarization has been demonstrated to drive early progressive tissue loss and dysfunction, serving as a good predictor of injury outcome following brain trauma [47,48]. Traditional stratification of monocytes includes classical (CD14^hi^CD16^-^ human; Ly6C^hi^CD43^-^ mouse), non-classical (CD14^dim^CD16^++^ human; Ly6C^lo^CD43^+^ mouse), and intermediate (CD14+CD16+; Ly6C^int^CD43^+^) subsets [10,49,50]. It is well established that these monocyte subsets infiltrate the brain following traumatic injury in mice and that the CCR2-CCL2 axis is essential for the recruitment of the Ly6C^hi^ subset [51]. Classical monocytes are involved in phagocytosis and pro-inflammatory responses; intermediate cells exclusively express *CCR5* and mediate antigen presentation, while non-classical cells are involved in complement and Fc gamma-mediated phagocytosis and adhesion [52,53,54]. A subset of non-classical monocytes express CD9 and SLAN, which suggests increased efferocytosis and migration functions compared to the SLAN-negative population [53,55]. Subsets of macrophages have been identified in the injured brain [48].

Monocyte depletion studies demonstrated that peripheral-derived monocytes contribute to the pathophysiology of several neurological disorders, including but not limited to Alzheimer’s disease (AD), intracerebral hemorrhage, traumatic brain injury, and multiple sclerosis [51,52,56,57,58,59,60,61,62,63,64,65]. Mice receiving intravenous injections of clodronate had reduced numbers of CCR2+Ly6c^hi^ monocytes infiltrating the brain following injury, suggesting subset control modulates tissue damage [52]. Importantly, the use of CCR2-/- mice revealed that altering monocyte influx directly affects the production of type1 interferon genes in resident microglial subsets [47]. Loss of Ccr2 also accelerates early disease progression, cognitive impairments, Aβ plaque burden and alters microglia accumulation and phenotype in murine models of Alzheimer’s disease [62,63]. These studies demonstrate the likely influence of peripheral-derived monocytes on microglia activation in the diseased brain. Bone marrow-derived macrophages (BMDMs) play crucial roles in the pathophysiology of numerous neurological conditions, and their modulation (through bone marrow transplantation or gene therapy) represents a key therapeutic strategy for treatment, as reviewed elsewhere [66].

### 1.3. CNS Border-Associated Macrophages

Single-cell RNA sequencing (scRNA-seq) of the immune compartments within and bordering the CNS identified microglia as the long-lived, yolk sac-derived myeloid cells within the CNS parenchyma, while border-associated macrophages (BAMs) were the primary myeloid cells within the peri-vascular (pvMɸ), choroid plexus (cpMɸ), and meninges (mMɸ) [67,68,69]. The central role of BAMs in normal brain physiology and the pathogenesis of disease is reviewed elsewhere [1,9]. Briefly, pvBAMs may contribute to vascular inflammation, blood–brain barrier (BBB) permeability, nutrient exchange, and metabolism, as well as HPA axis regulation [70,71]. Given that the choroid plexus interfaces between the periphery and CNS, immune cell activation in this compartment may aid in T-cell trafficking and antigen presentation in the meninges [67,68,72], along with neutrophil invasion [69,73]. With an opportune location, cpMɸ are crucial in the development of neurological sequalae of multiple sclerosis (MS), experimental autoimmune encephalomyelitis (EAE), and TBI. In meninges, the fenestrated endothelium grants passage of microorganisms to prime the brain against future potential harm from circulating microorganisms in the context of injury; this concept has been termed the “gut-meningeal immune axis” [74]. mMɸ are both physically and transcriptionally distinct from microglia [33], as demonstrated in mild TBI patients where meningeal vascular damage was resolved owing to local and infiltrating CD206^+^ myeloid cells that scavenged dead cells and promoted angiogenesis [75]. As the immune gatekeepers of the CNS, BAMs control over immune cell entry, CSF/blood exchange, and debris clean-up makes them key contributors to neuroinflammation in acquired neurological disease.


 **b.** 
**New insights into disease-associated CNS myeloid subsets**



### 1.4. CNS Resident Microglial Subsets

There is a transcriptional distinction between homeostatic resident microglia and their disease-associated counterparts [76,77]. The main classifications of microglia have been extensively reviewed [18,39,78]. Briefly, mature microglia display a homeostatic, surveilling phenotype, express key TFs *Jun*, *Fos*, *Mef2a*, and *Mafb*, and the characteristic markers TMEM119, purinergic and chemokine receptors P2ry12, Csf1R, and Cx3cr1. Csf1R, for example, is critical for microglial survival and its null mutation removes 99.7% of the microglial cell population, while a few morphologically-distinctive microglia remain near the piriform cortex, hippocampus, thalamus, and dentate gyrus [79]. Zhan L. et al. used *CX3CR1-CreERT2/Rosa26-stop-DsRed* mice to show that repopulation after administration of the Csf1R inhibitor, PLX5622 (PLX), [80,81] required a PLX-resistant, nestin-positive microglial progenitor that displays upregulation of *Galectin-3* or *Mac2* (a ligand for Trem2 [82]) and other immature microglial genes and downregulation of mature markers *Tmem119*, *Mafb*, *Cx3cr1,* and *Csf1r* [83]. Others showed that the *Itgax* (CD11c^+^) microglia subset is highly proliferative and prominent in the developing CNS in areas of myelination that express *Igf1, Spp1*, and *Gpnmb* [37,84].

While recent scRNAseq findings show several distinct homeostatic subtypes based on cluster analysis in the human and mouse brain [76,85], putative phenotypes with unique specifications, such as axonal interaction, supporting myelination and neurogenesis, and synaptic pruning, have been described in different brain regions [36]. However, following insult, it is well-established that microglia diversify their phenotypic identity and function, resulting in a continuum of states that include pro-inflammatory and/or anti-inflammatory signatures in response to environmental stimuli [18,21,30,42,86]. In general, their expression of pro-inflammatory cytokines, such as TNF, IL1β, IL12, and reactive oxygen and nitrogen species (ROS and RNS, respectively) can mediate the recruitment of innate immune cells and induce neurotoxic effects [14]. Microglia expression of nitric oxide synthase (iNOS) also contributes to the neuroinflammatory environment and neuronal cell death [25]. A review of traditional classifications of phenotypic states identified in brain injury has been discussed previously [29,42,87]. More recently, however, lineage tracing and scRNA-seq studies [3,8,76,85,88] suggest that more distinct subsets exist, highlighting a complex landscape in the context of neurological disease, where microglia may adopt both anti- [22] and pro-inflammatory features [48] with transcriptional states that vary based on brain region, age, and disease pathology [7,37,85,89]. Therefore, the traditional classifications [29,43,48] have been reconsidered, given the heterogeneity now observed across health and disease.

### 1.5. Subsets of Microglia in the Human Brain

scRNAseq and time-of-flight mass cytometry performed by Sankowski et al. revealed a spectrum of transcriptional states in human microglia during homeostasis, aging, and disease [35]. Nine clusters (C1–C9) were identified: C2-C3 were homeostatic clusters with high core gene expression *CX3CR1, CSF1R, P2RY12, P2RY13,* and *TMEM119*; C6–C7 clusters showed low CX3CR1, high MHC-II, and metabolism genes (such as *ApoE* and *LPL*); C1, 5, 8, 9 were characterized by high transcript expression of chemokines and cytokines. Importantly, cluster enrichment was region-specific with MHC-II^high^ clusters residing in white matter and MHC-II^low^ residing in gray matter microglia as well as age-specific with >50 years having enriched C6 and C7 with higher expression of *SPP1* (osteopontin), a pro-inflammatory cytokine. Similarly, following analysis of cortical microglia isolated from healthy human brains (with no evidence of CNS pathology), Masuda et al. identified four clusters of microglia (C1–C4). C1 and C2 had higher expression of *CST3* and *P2RY13* than C3 and C4. In addition, C4 had high levels of chemokine genes *CCL2* and *CCL4* and transcription factors *EGR2* and *EGR3,* indicating that homeostatic human microglia include subsets with distinct gene expression patterns [89].

Esaulova E. et al. surprisingly showed a discrete population of microglia having homeostatic genes *cx3cr1, csf1r, slc2a5, marcks,* and *P2ry13* in the CSF and blood of patients with MS. When applying the classifier tool from Sankowski et al., the cells were identified as microglia cluster six and seven with gene ontology described as antigen processing [88]. Microglia heterogeneity was further validated in the brains of patients with temporal lobe epilepsy (TLE), mild cognitive impairment (MCI), and Alzheimer’s disease (AD), showing nine distinct clusters that expressed microglia-enriched genes. Clusters one and two were the most common across all patients and were considered typical homeostatic states. C1–3 comprised cells with closely related signatures, as did C5 and C6, but the remaining microglial clusters showed more distinct signatures. Further, cluster inter-relatedness suggested that differentiation of each cluster emerged radially from a common cell fate. The temporal neocortex of TLE patients showed an increased frequency of three microglia clusters: C5 (CD83-positive cells enriched with transcription factors CREB and ATF and anti-inflammatory genes *IL4*, *IL10*, and *IL13*), C6 (CD83-positive cells enriched with anti-inflammatory genes), and C7 (CD74^hi^ cells enriched with antigen-presenting genes). However, the dorsolateral prefrontal cortex of AD patients showed reduced frequency of C7 microglia [90]. In addition, the brains of healthy humans and patients with MS showed seven myeloid clusters expressing microglia core genes [89]. C5, C6, and C7 microglia are entirely from healthy brains showing the highest expression of core genes. C4 clusters contained microglia from both healthy and MS brains showing reduced core gene expression and elevated CCL2, CCL4, EGR2, and other cytokine and chemokine genes. C2, C3, and C8 microglia were enriched in the brains of MS patients showing downregulation or an absence of core genes, including *TMEM119,* and the upregulation of *APOE* and *MAFP*. C3 microglia showed increased expression of MHCII-related molecules, indicating an immunomodulatory role of this microglial subset. A detailed summary of the alterations in gene expression between microglial clusters in healthy and diseased human brains can be found in Table 1.

### 1.6. Subsets of Microglia in the Murine Brain

Evaluations of microglia in the murine brain have also revealed a more complex cell state using scRNA-seq and clustering analysis, which demonstrated distinct microglial clusters across age and injury [85,89]. Hammond et al. showed nine clusters (C1–C9) that exist across age and injury with differential gene expression observed to be cluster-specific, except canonical genes *C1qa*, *Fcrls*, *P2ry12*, *Cx3cr1*, *Trem2,* which were expressed in most clusters and *C1qa*, *Fcrls*, *Trem2* expressed regardless of age or perturbation. Likewise, *F13a1*, *H2Aa*, *Ccr2*, *Lyve1,* and *Mgl2* were identified as monocyte/macrophage-specific [85]. Embryonic brains were enriched with C6 microglia expressing *MS4a7* and sharing a similar transcriptional profile with mature microglia, peripheral macrophages, and BAM suggesting an intermediate state that possibly downregulates macrophage-related genes upon entry into the brain. Interestingly, no specific clusters appeared with aging; instead, an expansion of microglia expressing inflammatory and interferon-responsive signals was seen. Similarly, Masuda et al. revealed ten clusters (C1–C10) with variable distribution, both spatially and temporally, across embryonic and post-natal development [89]. Greater heterogeneity in subtypes occurred during embryonic development (C1–C6) compared to the postnatal period (C7–C10), whose post-natal expression includes homeostatic genes *Tmem119*, *Selplg,* and *Slc2a5*. In this study, embryonic clusters showed high expression of lysosomal genes (e.g., *ctsb*, *ctsd*, *lamp1*) and the amyloid cell activation marker (*Apoe*). These distinct populations disappeared in post-natal brains. Additional clusters emerged after facial nerve axotomy (C11) and cuprizone treatment (C12–C13) in a time-dependent fashion. Disease-specific signatures revealed upregulation of demyelination, remyelination, and MHCII genes, along with downregulation of TMEM119. The variable distribution of clusters across brain regions suggests that the local microenvironment drives subtype needs fostering expeditious expansion, enhanced synaptic pruning, cell motility, or immunological needs of the brain [23,91,92,93].

Similar studies showed that TBI resulted in four distinct clusters over the course of progressive injury, each (C1–C4) related to host defense response (C1), synaptic potentiation (C2), lipid remodeling (C3), and membrane polarization (C4) [31]. In EAE, four disease-associated microglial clusters (DAMg 3–6) were observed, showing downregulation of canonical genes and upregulation of genes related to immune regulation, cell activation, proliferation, and chemokine/cytokine production [76]. Flow cytometric analysis can aid in the identification of pro- and anti-inflammatory DAMs in AD models leading to studies on the response to various drug therapeutics. For example, the use of the Kv1.3 channel blocker, ShK223, was found to reduce the expression of pro-inflammatory while also influencing the production of anti-inflammatory DAM genes, phagocytic uptake, and clearance of AB [94]. The discovery of DAMs has created an opportunity to develop more targeted therapy aimed at modulating this inflammatory subset to quell neuronal injury in diseases of the nervous system. Collectively, these studies have advanced our understanding of the multi-faceted nature of microglial subpopulations which offer potentially innovative approaches designed to target defined subsets for therapeutic intervention. A summary of genes altered in murine microglia clusters within healthy and diseased brains is shown in Table 1.

### 1.7. CNS Border-Associated and Peripheral-Derived Macrophage Subsets

Each region-specific BAM population is endowed with distinct transcriptomic clustering, which has been shown to mediate immune responses at the brain boundaries and whose generation requires the transcription factor (TF) *PU.1*, unlike short-lived peripheral-derived myeloid cells such as monocytes and dendritic cells [7,76,95,96]. High-dimensional single-cell proteome mass cytometry shows that surface marker expression of CD38 and MHC-II are distinguishing features of BAMs at steady state, and their four subsets include Cd38^+^/MHCII^−^/Ccr2^−^, Cd38^+^/MHCII^+^/Ccr2^−^, Cd38^-^/MHCII^+^/Ccr2^−^, Cd38^-^/MHCII^+^/Ccr2^+^ [8]. Only Ccr2^+^ BAMs are enriched in the choroid plexus and are proposed to be replaced routinely by BM-derived monocytes [7,8]; these homeostatic phenotypes were also identified by scRNA-seq [76].

Multiparametric single-cell analyses have revealed subsets that represent either distinct stages of linear differentiation or functionally distinct subsets that emerge under emergency conditions, such as neutrophil-like monocytes [97], to initiate inflammation or promote healing [50]. Recent studies using Ccr2-CreER-based fate-mapping showed that monocytes infiltrating the brain after hypoxic-ischemic (HI) injury were maintained as TNF/MHC-II macrophages while others were Tmem119/Sall1/P2ry12 ramified microglial-like cells that persisted for months [17]. In addition, monocyte-derived macrophages were analyzed by scRNA-seq profiling in TBI and revealed a mixture of polarized subsets [98]. However, this may reflect the early stage (one-day post-injury), where the initial wave of immune cell entry may be from one subpopulation. These and other findings discussed later highlight the ability of peripheral-derived monocytes to generate several subsets, including those with microglial-like phenotypes and functions in the brain.


 **c.** 
**Peripheral-derived monocyte engraftment and cross-talk with microglia and BAMs**



Under defined experimental conditions, peripheral-derived macrophages have demonstrated a unique ability to replace the long-lived BAMs and microglia during neuroinflammation [99,100,101]. It has been demonstrated, using transcriptomic and proteomic analysis, that these populations have distinct regional ontogeny with divergent transcriptional and immunological signatures both in the development and diseases of the brain [102]. A number of previously identified microglial-specific markers, such as TMEM119, show reduced immunoreactivity in disease [103,104] and are expressed by peripheral-derived cells that enter the brain under pathological conditions such as ischemic stroke, EAE, MS, 6-OHDA [105,106,107], which we further demonstrated in our own work following focal traumatic cortical impact injury [108]. Furthermore, peripheral-derived macrophages have been shown to express transcripts for common microglia-specific markers early during development, including TMEM119 as well as Fcrls, P2ry12, and Trem2 [109]. Fate mapping experiments with reporter-labeling models have been instrumental in identifying cell origins and cross-talk between populations [15,95]; these include GFP bone marrow chimeras, and Cx3cr1^GFP^/Ccr2^RFP^, Cx3cr1^CreER^/R26^tdTomato/stopRFP^ mice (Figure 1).

### 1.8. Blood–Brain Barrier Integrity and Peripheral-Derived Immune Cell Invasion

In non-pathologic states, the brain parenchyma is a highly immune-privileged tissue separated from the blood via the BBB, a non-fenestrated endothelial barrier with tight junctions composed of transmembrane proteins such as claudins, occludins, and junctional adhesion molecules[110]. However, in the absence of neuroinflammation, perivascular BAMs and microglia interact with peripheral-derived T-cells via antigen presentation to facilitate immune surveillance [111]. Activated T-cells then cross the BBB endothelium via paracellular (tight junction-dependent) or transcellular (endothelial cell receptor and charge-dependent) transport mechanisms while preserving the integrity of the BBB [112,113]. As neuroinflammation progresses, chemoattractants (e.g., chemokine-like factor-1) mediate neutrophil migration, receptors (e.g., CCR2) recruit monocytes, and endothelial cell receptor expression (P-selectin, ICAM-1) promotes paracellular permeability, causing blood-borne immune cell infiltration into a disrupted BBB [114].

### 1.9. External Factors Influencing CNS Engraftment of Peripheral-Derived Monocytes

Peripheral-derived monocytes that infiltrate the CNS can acquire a microglial-like phenotype and function that is driven by environmental signals [101,115,116]. Under defined conditions, peripheral-derived monocytes/macrophages give rise to microglia that are phenotypically indistinguishable [101,115,116]. The microglia-vacant niche is replaced, in part, by peripheral-derived monocytes following lethal whole-body Ɣ irradiation in response to microglia depletion using the pharmacological agent PLX5622, genetic modification using *CD11b-HSVTKI*, and in tamoxifen-treated *Cx3cr1^CreER/+^/R26^DTA/+^* or *Cx3cr1^CreER/+^/Csf1r^f/f^* mice (Figure 2) [81,117]. The BBB is not disrupted, and microglia replacement requires the sustained engraftment of the Ly6c^hi^/Ccr2^+^ subpopulation that gives rise to F4/80^hi^ macrophages [99,115,117]. Long-term engraftment did not occur when the brain was shielded from irradiation or in control *Cx3cr1^CreER/+^* mice. Increased cytokine production is postulated as a key attractant of Ly6c^hi^ monocytes [51]. While these cells upregulate microglial-specific genes (*TMEM119*, *P2ry12*, *Siglech*, etc.), they remain transcriptionally independent from naïve microglia, lacking *Sall1* transcription factor but notably, upregulating transendothelial migration genes (*Itga4*, *Vcam1*, *Itgal*), which correlated with vascular activation suggesting that they gain entry through transcytosis mechanisms and have distinct morphology and function such as higher motility in response to stimuli [117]. This may be due to their unique epigenetic programming from the origin of birth. It should be noted that the source of irradiation, gamma vs. X-ray, and shielding from irradiation differs across studies. Microglia replacement has not been demonstrated using X-ray irradiation [7,52,118] but shows extensive repopulation in gamma-irradiated conditions [101,115]. Due to the recent discovery of a skull bone marrow and myeloid cell reservoir, it is plausible that engraftment may also occur as a result of this population [11,119,120,121]. However, parabiotic experiments [115] and head-shielded chimeras using CD45.1 and CD45.2 cells [117] suggest that the CNS-engrafted cells are primarily from circulation.

The loss of Cx3cr1^hi^Ccr2^lo-neg^ meningeal macrophages after mild TBI can also result in the invasion of Cx3cr1^lo^Ccr2^hi^ monocytes; however, this response is transient and temporally regulated [75]. Studies have shown that BAMs are readily and almost completely exchanged after artificial whole-body irradiation and bone marrow transplantation [122,123]. Meningeal macrophages (mMɸ) serve as sentinels for foreign invaders and are gatekeepers of the neuroimmune response. Peripheral-derived monocytes can also repopulate an open niche created by meningeal macrophage loss, for example, by viruses; however, they fail to fully mimic their predecessors [116,124]. Lineage tracing via Cx3cr1^CreER^/Stop^fl/fl^ TdTomato mice showed that viral destruction of mMɸ resulted in their repopulation by circulating monocytes whose gene profile was enriched for IFN-Ɣ-related genes paired with a lower frequency of MHC-II. This event appears to be context-dependent as additional *Kit*^MerCreMer^/*R26*^YFP^ fate-mapping analysis [125,126] has demonstrated that microglia and F4/80^hi^CD206^+^ CNS border-associated macrophages were stable during the progression of AD pathology whose turnover rate by circulating monocytes was less than 10% (Figure 2) [126].

### 1.10. Maintenance of Distinct Identities Following Cell Replacement

Using fate tracing methods to discriminate BAMs, microglia, and PDMs whole genome sequencing (WGS) has identified several specific signatures. Cronk J. et al. identified 52 microglia genes that were not present in brain-engrafted peripheral-derived macrophages [115]. Grassivaro F. et al. identified 65 unique transcripts that were distinct throughout the life of microglia that were not expressed by peripheral macrophages [109]. When comparing these lists, we observed that *sparc*, *capn3*, *rtn4rl1*, *sall1*, and *sall3* were unique to microglia regardless of myeloid comparison. The use of *Sall1^GFP^* reporter mice confirmed that this transcription factor is restricted to microglia and not BAM or peripheral-derived macrophages [8]. The Spalt-like transcription factors *sall1* and *sall3* were also noted to be microglia-specific when comparing the competitive repopulation after microglia depletion [117] and using gene expression profiling [127,128,129]. *Sall1*-deficient mice display proinflammatory, ameboid microglia [129], altered neurogenesis, and tissue homeostasis [127]. *Sall1* is also downregulated in microglia following intracerebral hemorrhage (ICH) at 1 day but is restored at later time points suggesting that suppressing its expression mediates the pro-inflammatory state in the acute phase of disease. Among the core microglial genes, the *Sparc* transcript has been shown to be stable during EAE pathology [76] but is rapidly downregulated at the protein level in ischemic and excitotoxic lesions, mediating microglia proliferation [130]. These studies demonstrate that while peripheral-derived macrophages can act as a cell source for microglial replacement (due to damage or senescence), microglia remain transcriptionally distinct from other myeloid cell subsets.


 **d.** 
**Divergent neuroinflammatory functions of CNS macrophages in diseases of the brain**



Peripheral-derived myeloid cells gain access to the brain in the presence of an intact BBB due to the local inflammatory environment created by experimental elimination of microglia following gamma irradiation or depletion. Their recruitment and coordination with resident microglia in mediating neuroinflammation as a consequence of acquired injury such as stroke, TBI, AD, and cancer (glioblastoma, gliomas) have been extensively evaluated [5,12,14,21,40,63,90,131,132,133,134,135,136,137].

Early clonal expansion of resident microglia occurs alongside infiltrating monocytes/macrophages, which transiently integrate into the parenchyma following EAE [76]. Using single-cell profiling in this same model, distinct myeloid fates were identified in the neuroinflammatory milieu, where microglia generate defined subsets while their role in antigen presentation is co-opted by dendritic and monocyte-derived cells with prolonged T-cell interactions, which are presumed to be the major players in disease progression [88]. This comports with additional findings that show higher expression of MHCII (*H2aa*, *H2ab1,* and *Cd74*), as well as more efficient antigen presentation and T-cell proliferation in peripheral-derived macrophages compared to microglia in the hemorrhagic brain [102]. Transcriptional profiles of resident microglia and CNS-infiltrating macrophages showed enriched gene sets that remained distinct over time post-injury, with microglia displaying genes necessary for immune migration, complement (*C1qb*, *C1qa*, *Trem2* for synaptic pruning), and inflammation [3,8,76,89]. Subsets of disease-specific myeloid cells also can be identified in the brain and pia following ischemic stroke. Single-cell transcriptomic analysis described four microglia clusters with *Tmem119*, *Hexb*, *Cx3cr1*, *Sparc*, *P2ry12*, *and Cst3* as transcript signatures and five macrophage clusters with *CAM*, *SAMC*, *MARCO*, *Lyz2,* and *Apoe* transcripts, one of which showing BAM expression of *Pf4 and Lyve1* [138]. These studies suggest that, while resident and CNS-infiltrating macrophages co-exist in the diseased brain, they play complementary roles that are not mutually exclusive in neuroinflammation.

## 2. Clinical Assessment of Microglial Activation and CNS-Infiltrating Macrophages

Infiltration of peripheral-derived monocyte/macrophages, along with activation of resident and CNS-associated macrophages, play a significant role in the progressive pathology of brain disorders such as acquired injury and degenerative conditions. A number of clinical approaches have emerged to evaluate these changes using CT, PET, and MRI imaging. To improve clinical management across a variety of brain disorders, monitoring of microglial activation in vivo will aid in the selection and implementation of novel therapeutics. Several studies have demonstrated that new imaging techniques offer direct visualization of neuroinflammatory sequelae. When these imaging techniques are combined with microglial-specific biomarkers and clinical assessments, their value in predicting disease outcomes may be greater than alone. As disease-associated microglia and CNS or peripheral macrophages orchestrate the progression of the CNS inflammatory response, advanced imaging techniques in the clinical setting to monitor these myeloid changes may aid in developing prognostic biomarkers for neurological disorders.


 **a.** 
**Imaging techniques: Positron emission tomography (PET), diffuse-weighted MRI, and Magnetic Resonance Imaging (MRI)**



To date, several microglia-related PET radiotracers and MRI contrast agents have been administered to patients with AD, IS, and TBI to assess the extent of microglia-mediated neuroinflammation. Overall, findings utilizing these new modalities suggest that cell-specific neuroinflammation can be visualized with whole-brain imaging. Translocator protein 18 kDa (TSPO), a benzodiazepine receptor expressed in the mitochondrial outer layer of microglia, is a target in PET imaging (Yao et al., 2020). One TSPO-targeting radioligand, 1-(2-chlorophenyl)-N-[11C]methyl-N-(1-methylpropyl)-3-isoquinolinecarboxamide ([11C]PK11195), is lipophilic and exhibits non-specific binding but its *R*-enantiomer binds TSPO with 2x greater affinity [139] and demonstrates comparable levels in amyloid load in the brains of AD patients [140]. In TBI, [11C]DAA1106 is highly selective for TSPO and localizes to the region of injury for a longer time than the [11C]PK11195 *R*-enantiomer [141]. TSPO is also expressed in blood-borne infiltrating monocytes/macrophages, suggesting that this radioligand target may be reflective of broad myeloid activation [142]. To mitigate this limitation of PET imaging, a recent pre-clinical study investigated the utility of diffusion-weighted MRI to map microglia activation [143]. This method is specific to microglia through the combination of three MRI parameters: the stick fraction, the stick dispersion, and the small sphere size. Lastly, MRI imaging of blood-borne macrophage invasion across several disease states is possible when using ultra-small particles of iron oxide (USPIO) [144]. The combination of TSPO-targeting PET (local microglia and blood-borne macrophages) and iron oxide-enhanced MRI (blood-borne macrophages) may elucidate the extent of microglial vs. peripheral-derived mediated changes.


 **b.** 
**CSF and blood biomarkers to detect microglia activation**



While MRI and PET may detect microglia activation in vivo, limitations such as cell specificity and radioligand quality require the inclusion of additional assessments such as soluble biomarkers present in blood or CSF. There is a growing need to develop biomarkers to detect microglia subsets in the clinical setting. To date, several microglia-related biomarkers have been measured in CSF and serum of patients with neuroinflammation. These are summarized in Table 2.

These markers include triggering receptor (*TREM2*) expressed on myeloid cells [145,146] whose cleaved, soluble form (sTREM2) is detectable in CSF and may reflect the microglial response, especially in AD, sports-related repetitive TBI, ischemic stroke, and MS [135,145,146,147,148,149,150,151,152,153,154]. *NLRP3 inflammasome* [155,156,157,158] is increased in CSF and associated with worse functional outcomes in pediatric TBI patients [159]. *Adiponectin*, an adipocyte-secreted protein signal [160], is also increased under a prolonged state of neuroinflammation in AD [161], is used as a diagnostic tool in severe TBI [162], and associated with disease severity in MS [163]. Finally, *high mobility group box protein 1 (HMGB1)* is highly expressed in phagocytic microglia [164], which may reflect AD progression [165,166] as well as ischemic stroke, TBI, and MS severity [167,168,169,170]. *Galectin-3* drives microglial activation [171,172,173,174] and is associated with a worse AD [175,176], TBI, and stroke [173,177,178]. *Fractalkine/CX3 chemokine ligand 1 (CX3CL1)* is higher in the plasma of patients with lower stroke severity [179]. *Progranulin (PGRN)* may attenuate aberrant microglia/macrophage-mediated neuronal damage [180,181,182] and its diagnostic utility in AD is sex- and age-dependent [147,183]. Higher serum levels of *GPRN* are associated with a high risk of adverse functional outcomes in stroke and differentiate radiologically isolated MS from healthy controls [184,185]. Finally, the *transmembrane Protein 119 (TMEM119)* could be a biomarker of AD progression [186].

A combinatory approach that includes advanced imaging and blood/CSF biomarkers reflecting microglial changes, along with clinical assessments, may have greater predictive value in monitoring disease progression. Recently it was shown that combining (non-microglial) MRI, PET, CSF biomarkers, and clinical assessment better predicted a patient’s conversion to AD than a clinical assessment alone [187]. Thus, we propose that the development of a suitable pipeline that includes microglia-specific imaging, selected biomarkers, and clinical assessment may more accurately predict disease outcomes than each of these approaches alone.
cells-11-03364-t002_Table 2Table 2Blood/CSF biomarkers targeting microglia in AD, IS, TBI, and MS.Biomarker NameAbbreviationMicroglia ConnectionDisease StateSourceDirectionSourcesSoluble triggering receptor expressed on myeloid cells 2sTREM2Expressed in microglia, increases inflammatory cytokinesADCSFUp[135,145,147]ISBloodUp[149,150,151]TBICSFUp[148]MSCSFUp[152,153,154]Nod-leucine rich repeat and pyrin containing protein 3NLRP3 inflammasomeActivated by microglial cytokines IL-1β and IL-18ADCSFUp[41,155]ISBrain tissueUp[156]TBICSFUp[159]MSPBMCUp[158]AdiponectinAdiponectinModulates microglia through PPAR-γ signalingADCSF/BloodDown/Up[161]ISBloodUp[188]TBIBloodUp[162]MSBloodUp[163]High mobility group box protein 1HMGB1DAMP, expressed in phagocytic microgliaADBloodUp[165,166]ISBloodUp[167,168]TBIBloodUp[169]MSBloodUp[170]Galectin-3Gal-3TLR-4 ligand, promotes microglial activationADBloodUp[175,176]ISBloodUp[173,177]TBIBloodUp[178]MSBlood/AAUp[174]FractalkineCX3CL1Expressed in neurons, facilitates neuron-microglia communicationADCSF/BloodDown/-[136]ISBloodDown[179]TBIBlood/CMDMixed[40,189]MSPBMCUp[190]ProgranulinPGRNExpressed in microglia, attenuates neural damageADCSFUp, sex/age-bias[183]ISBloodUp[184]TBISerumMixed[180,191]MSCSFUp[185]AD = Alzheimer’s disease; IS = Ischemic stroke; TBI = Traumatic brain injury; MS = Multiple sclerosis; CSF = Cerebral spinal fluid; PBMC = Peripheral blood monocytes; AA = Autoantibodies; CMD = Cerebral microdialysis; DAMP = damage associated protein.

## 3. Summary

Neuroinflammation caused by disease, or through experimental strategies to deplete long-lived resident microglia, can allow entry of peripheral-derived monocytes/macrophages (PDMs) into the brain. This may act as a cell source for microglia replacement or as a partner in mediating the neuroinflammatory milieu. Recently it was shown that regardless of how many PDMs entered the brain, their polarization state was most important in dictating the outcome of brain injury. It remains unresolved which transcriptionally distinct PDM subsets influence early vs. chronic stages of disease and whether they work in concert with resident microglia to restore homeostatic balance. Importantly, future studies may address sex differences and how PDM subsets interact with ontologically distinct microglia to regulate their phenotypic state and whether this cross-talk can be recalibrated to control the expression of pro- and anti-inflammatory responses. Overall, resident microglia are key players in the progression and resolution of injury to the CNS. Microglia production of inflammatory cues aids in the recruitment of peripheral-derived immune cells, including monocyte/macrophages, whose coordination may be central to regulating the neuroinflammatory milieu across numerous neurological disorders. The use of single-cell RNA sequencing has helped detail the complex transcriptomic characteristics of newly identified myeloid derivatives, as well as enhanced our understanding of their origins and local responsibilities across the lifespan.

Lastly, there is a need to develop new combinatory diagnostic tools (blood-based and/or imaging biomarkers) in the clinical setting to track microglial/PDM activation states in the brain. This advancement would improve prognostic indicators of disease outcome and enable the successful monitoring of new therapeutic strategies aimed at quelling a progressive pro-inflammatory response. Using murine models, we understand that PDMs can enter the brain in the absence of BBB disruption through the upregulation of genes required for transcytosis. The identification of pro-resolving PDM subsets that are long-lived in the brain under disease conditions may be an attractive cell therapy to combat chronic neuroinflammation.

## Figures and Tables

**Figure 1 cells-11-03364-f001:**
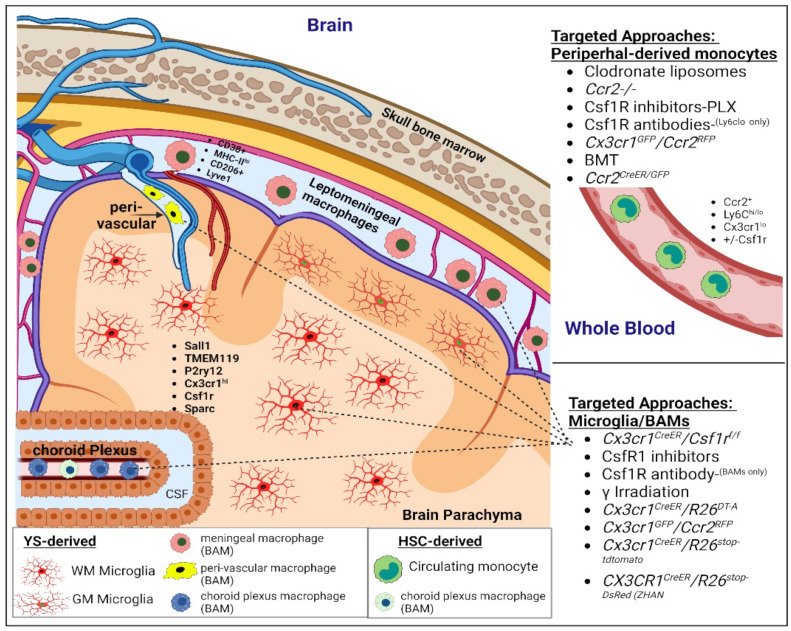
Peripheral-derived monocytes, microglia, and CNS border-associated macrophages (BAMs). Regional presence and experimental approaches used to target CNS macrophages vs. circulating monocytes. Lineage tracing and scRNAseq have been instrumental in understanding the phenotypic spectrum and subpopulations of resident vs. CNS-infiltrating macrophages. (YS—yolk-sac, BAM—border-associated macrophages, GMؙ—grey matter, WM—white matter, HSC—hematopoietic stem cells, BMT—bone marrow transplant).

**Figure 2 cells-11-03364-f002:**
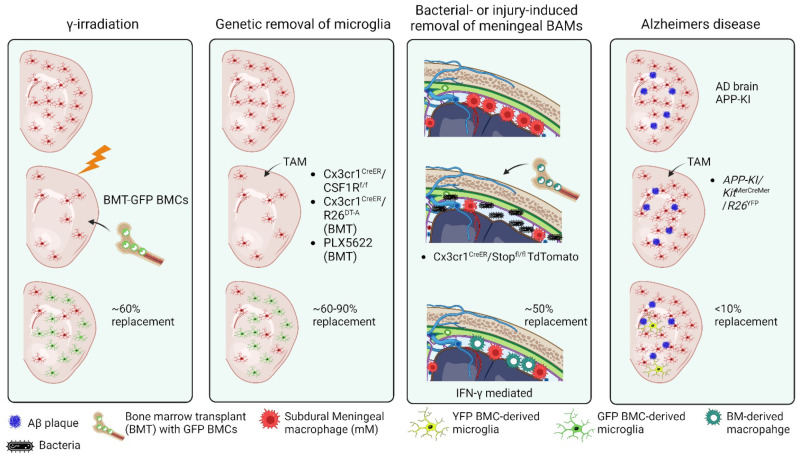
Peripheral-derived monocytes replace microglia and BAMs under defined experimental conditions, infection, and Alzheimer’s disease. Panel 1: gamma irradiation-induced replacement; Panel 2: genetic or pharmacological methods to delete microglia induces replacement by peripheral cells; Panel 3: Bacterial infection or brain injury loss of border-associated macrophages (BAMs) induces replacement by peripheral cells; Panel 4: Alzheimer’s disease induces minimal replacement of microglia by peripheral myeloid cells. BMT = bone marrow transplant; AD = Alzheimer’s disease; TAM = tamoxifen; BMCs = bone marrow cells; YFP = yellow fluorescent protein; GFP = green fluorescent protein.

**Table 1 cells-11-03364-t001:** Human gene signatures of microglia subsets in healthy and diseased brain. Microglia clusters identified by the expression of core genes.

Study	Human Tissue	Techniques	Subsets/Clusters	Enriched Genes	Function
[35]	Healthy tissue Temporal lobe (Grey & white matter)	scRNAseq Time-of-flight mass cytometry Cross-data comparison	9 clusters (C1–C9)		
			C1	*CCL2, IL1B*	Chemokine and cytokine inflammatory genes
			C2 (WM)	*MHC-II, HLA-DRA, CD74 IFI44L*	Chemokine and cytokine inflammatory genes
			C3 (GM)	*CX3CR1, TMEM119*	Homeostasis c
			C5 (WM)	*MHC-II,CCL2 IL1B*	Homeostasis
			C8 (GM)	*CCL2, IL1B*	Chemokine and cytokine inflammatory genes
			C6–C7 (WM)	*MHC-II, SPP1, APOE (>50 y) and LPL*	Integrin-receptor-binding protein and metabolism genes
[89]	Healthy tissue MS tissue	scRNA -seq (Cel-Seq2 protocol)	4 clusters (C1–C4)		
			C1–C2	*CST3, P2RY13*	Microglia activation and homeostasis
			C4	*CCL4, CCL2, EGR2, EGR3*	Cytokine inflammatory genes and zinc finger transcription factors
			7 clusters (C2–C8)		
			C2	*APOE, MAFB,* *CTSD, APOC1,* *GPNMB,* *ANXA2, LGALS1*	Microglia activation
			C3	*APOE, MAFB,* *CD74,* *HLA-DRA,* *HLA-DRB1* *HLA-DPB1*	Microglia Activation & Immunoregulation
			C4	*CCL2, CCL4,* *EGR4*	Cytokines and zinc finger TF
			C5–C7	*TMEM119, P2RY12*	Homeostasis
			C8	*APOE, MAFB,* *SPP1,* *PADI2, LPL*	Microglial Activation & Demyelination
[90]	TLE, MCI, AD	scRNA-seq	9 clusters (C1–C9)		
			C4 (MS, AD)	*IRF1, IRF8,* *IFITM3, ISG15+*	IFN-response
			C5 (TLE)	*CREB, ATF, IL10,* *IL4, IL13, CD83*	TF & anti-inflammatory
			C6 (TLE)	*IL10, IL4,* *IL13, CD83+*	Anti-inflammatory
			C7 (TLE, MCI; reduced in AD)	*APOE, TREM2,* *CD74^hi^*	Microglia activation and Antigen presentation
			C9	*E2F1, CBFB, NRF1*	Cell cycle

WM = white matter; GM = grey matter; TLE = temporal lobe epilepsy; MCI = malignant cerebral infarction; scRNA-seq = single cell RNA sequencing.

## Data Availability

Not applicable.

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
