# Peer review of "Cross-Talk and Subset Control of Microglia and Associated Myeloid Cells in Neurological Disorders"

_cells, 2022, doi:10.3390/cells11213364_

Round 1

Reviewer 1 Report

Dear Editor,

The review paper “Crosstalk and subset control of microglia and associated myeloid cells in neurological disorders of the brain” provides a detailed overview and most recent evidence gathered from sn-RNA-Seq and other high-throughput approaches of the role of microglia in the pathogenesis of CNS disorders.

The authors describe the differences found across several microglial subsets obtained from rodents, humans and models of neurological diseases/trauma. They also report how spatio-temporal factors (if the tissue used for microglial characterisation was taken from a damaged area of from a neighbouring compartment and when) can influence the clustering of these microglial subsets.

Another important topic detailed in this work is the emerging role of border-associated macrophages (BAMs) in replacing resident microglia when these are depleted (either experimentally or following certain CNS conditions).

Finally, the study reports the numerous attempts to identify and use molecules to capture microglial populations using different imaging modalities (i.e. TSPO in PET studies and USPIO in MRI), as well as the most recent microglial biomarkers reported in CSF and blood samples (elegantly summarised in Table 2).

The review is well-written and report the most up-to-date evidence on the role of microglia in the healthy and diseased CNS. This reviewer believes that the manuscript only requires some minor edits, which can help to further improve the quality of the current submission.

Minor criticisms

1.       At least 167 in-text references found but only 92 are listed in the reference list. Please make sure the list is comprehensive and all references are correctly listed.

2.       Page 8, line 244 and 258 – Same title and subtitle. Please replace the subtitle.

3.     The paper provides a comprehensive analysis of the role of microglia in traumatic (TBI), ischaemic (stroke) and neurodegenerative diseases (AD). However, in view of the strong neuroinflammatory nature of multiple sclerosis and contribution of microglia in MS pathogenesis, it would be nice if the authors could expand the two final sections on Imaging techniques and CSF/Blood markers to also include this neuroinflammatory condition. Accordingly, Table 2 would also need to be updated.  

The main reason for this recommendation is that many of the identified blood/CSF microglial biomarkers lack specificity towards the disease. For instance, sTREM2 has been recently found to be also upregulated in the CSF of people with MS (PMID: 35382840) along with neurofilament light chain (PMID: 35241571), whereas IL-1 receptor antagonist in CSF seems to be more specific for MS pathology given its correlation with disease score in relapsing-remitting MS (PMID: 34004435).

Author Response

Comment 1: At least 167 in-text references found but only 92 are listed in the reference list. Please make sure the list is comprehensive and all references are correctly listed.

Response: We appreciate the reviewer pointing out this inconsistency. It appears that our EndNote tool improperly synced when importing citations, resulting in this discrepancy in citations. We have re-synced our paper with EndNote and the reference list now has the appropriate 167 references.

Comment 2: Page 8, line 244 and 258 – Same title and subtitle. Please replace the subtitle.

Response: Thank you for this comment. We have updated the title and subtitle. “Peripheral-derived monocyte engraftment and cross-talk with microglia and BAMs” and External factors influencing CNS engraftment of peripheral-derived monocytes”

Comment 3:  The paper provides a comprehensive analysis of the role of microglia in traumatic (TBI), ischaemic (stroke) and neurodegenerative diseases (AD). However, in view of the strong neuroinflammatory nature of multiple sclerosis and contribution of microglia in MS pathogenesis, it would be nice if the authors could expand the two final sections on Imaging techniques and CSF/Blood markers to also include this neuroinflammatory condition. Accordingly, Table 2 would also need to be updated. 

The main reason for this recommendation is that many of the identified blood/CSF microglial biomarkers lack specificity towards the disease. For instance, sTREM2 has been recently found to be also upregulated in the CSF of people with MS (PMID: 35382840) along with neurofilament light chain (PMID: 35241571), whereas IL-1 receptor antagonist in CSF seems to be more specific for MS pathology given its correlation with disease score in relapsing-remitting MS (PMID: 34004435).

            Response: We greatly appreciate this comment and agree with the critique. We have updated our table 2 to add MS studies investigating the biomarkers already covered; thus PMID 35382840 has been included in this discussion. See table 2 and section 2b in the text for an updated discussion of prognostic biomarkers across neurologic diseases, including MS.

Reviewer 2 Report

In this review paper (cells-1934385), the authors discussed the role of microglia and associated  myeloid cells in neurological disorders. This topic is hot. Some concerns and suggestions are listed as below:

The authors said that three monocyte subsets infiltrate the brain following traumatic injury in mice and that the CCR2-CCL2 axis is essential for the recruitment of the Ly6C hi subset. The role of three subsets of monocytes in brain injury should be discussed separately.

In line 73, what do you mean by saying 'the production of several neurological disorders'?

The authors said that mice receiving systemic injection of clodronate had reduced numbers of CCR2+Ly6chi monocytes infiltrating the brain following injury. However, liposomal clodronate injection can be regarded as another method to specifically deplete microglia (PMID: 28629387). This effect should not be ignored.

Emerging data has convincingly demonstrated the existence of sex-dependent structural and functional differences of microglia, consequently changing our current understanding of these versatile cells (PMID: 33731174). This point should be discussed in the conditions of neurological diseases.

Accumulating evidence suggests that disease-associated microglia (DAM), a recently identified subset of CNS resident macrophages found at sites of neurodegeneration, might play such a protective role. The discovery of DAM created an opportunity to develop a therapy targeting the universal and intrinsic mechanism of fighting against neuronal death shared across multiple neurodegenerative conditions. This point should be discussed.

In research settings, microglia have to be dissociated from the CNS into a single-cell suspension using both mechanical dissociation and enzymatic dissociation in order to facilitate culture or analyses. Some may argue that mechanical dissociation and enzymatic dissociation may alter the microglial activation status and mRNA expression profiles.

Circulating monocytes are precursors of both tissue macrophages and dendritic cells, and they can infiltrate the central nervous system (CNS) where they transform into bone marrow-derived macrophages (BMDMs). BMDMs play essential roles in various CNS diseases, thus modulating BMDMs might be a way to treat these disorders. Understanding the distinct roles that BMDMs play in CNS diseases and their potential as gene delivery vehicles may provide new insights and opportunities for using BMDMs as therapeutic targets or delivery vehicles (PMID: 36066198). This point should be discussed.

No need to say neurological disorders of the brain. 'Neurological disorders' is fine.

Author Response

Comment 1: The authors said that three monocyte subsets infiltrate the brain following traumatic injury in mice and that the CCR2-CCL2 axis is essential for the recruitment of the Ly6C hi subset. The role of three subsets of monocytes in brain injury should be discussed separately.

            Response: We thank the reviewer for this suggestion.  We have expanded content of the subsets including their role in injury: “Classical monocytes are involved in phagocytosis, pro-inflammatory responses, intermediate cells exclusively express CCR5 and mediates antigen presentation, while non-classical cells are involved in complement and Fc gamma-mediated phagocytosis and adhesion [50-52]. A subset of non-classical monocytes express CD9 and SLAN which suggests increased efferocytosis and migration functions compared to the SLAN- population [51, 53].” And we add additional content in 74-78 regarding their role in injury including subset control examples.

Comment 2:  In line 73, what do you mean by saying 'the production of several neurological disorders'?

            Response: Thank you for pointing out this semantic error.  We have replaced ‘production’ with ‘pathophysiology’.     

Comment 3: The authors said that mice receiving systemic injection of clodronate had reduced numbers of CCR2+Ly6chi monocytes infiltrating the brain following injury. However, liposomal clodronate injection can be regarded as another method to specifically deplete microglia (PMID: 28629387). This effect should not be ignored.

            Response: Thank you for pointing out this method of clodronate depletion of microglia.  We would like to note that the paragraph details the depletion of only peripheral immune cells by way of intravenous injection via tail-vein prior to injury.  It should also be noted that clodronate is unable to cross the BBB on its own and that direct hippocampal injection is needed to deplete neural resident microglia (PMID: 28629387 & 28629387).  We have further refined the text to include method of intravenous injection.

Comment 4: Emerging data has convincingly demonstrated the existence of sex-dependent structural and functional differences of microglia, consequently changing our current understanding of these versatile cells (PMID: 33731174). This point should be discussed in the conditions of neurological diseases.

            Response: We thank the reviewer for this comment and agree there is room for expanding studies on sex differences. Because this review highlights the new findings of single cell Seq and the cross-talk between myeloid compartments, few studies address this new direction in research across the sexes. Until such time, we believe the focused review on myeloid crosstalk provides suitable updates in the field. The need for these studies in now included in the summary of the review.

Comment 5: Accumulating evidence suggests that disease-associated microglia (DAM), a recently identified subset of CNS resident macrophages found at sites of neurodegeneration, might play such a protective role. The discovery of DAM created an opportunity to develop a therapy targeting the universal and intrinsic mechanism of fighting against neuronal death shared across multiple neurodegenerative conditions. This point should be discussed.

            Response: We thank the reviewer for this thoughtful comment. Line 226; PMID: 29784049. We have chosen to elaborate further into these findings by noting a pathway used to influence the increase of anti-inflammatory DAMs and their potential for therapeutic targeting.

Comment 6: In research settings, microglia have to be dissociated from the CNS into a single-cell suspension using both mechanical dissociation and enzymatic dissociation in order to facilitate culture or analyses. Some may argue that mechanical dissociation and enzymatic dissociation may alter the microglial activation status and mRNA expression profiles.

            Response: We thank the reviewer for noting this important methodological difference. We would note that recent studies have found that although enzymatic dissociation can affect resident immune cells of the CNS, those affected are mostly astrocytes and not microglial cells. (PMID: 21989594 & 30559476). Cluster analysis also appears very consistent across studies using divergent techniques so it less likely to play a decisive role in the general findings in the field. We greatly appreciate mention of this technical challenge.

Comment 7: Circulating monocytes are precursors of both tissue macrophages and dendritic cells, and they can infiltrate the central nervous system (CNS) where they transform into bone marrow-derived macrophages (BMDMs). BMDMs play essential roles in various CNS diseases, thus modulating BMDMs might be a way to treat these disorders. Understanding the distinct roles that BMDMs play in CNS diseases and their potential as gene delivery vehicles may provide new insights and opportunities for using BMDMs as therapeutic targets or delivery vehicles (PMID: 36066198). This point should be discussed. (Jatia)

            Response: We agree that this population offers great opportunity for therapeutic targeting as a means to alter the environmental milieu in which the infiltrate. Additional comments on the therapeutic potential of modulating BMDMs has been added to the section on peripheral-derived monocytes, line 80 including a reference to the extensive review previously published on the topic.

Comment 8: No need to say neurological disorders of the brain. 'Neurological disorders' is fine.

            Response: We thank the reviewer for this suggestion. We have updated our title, line 11, line 28, and line 342 to say “neurological disorders” and not “neurological disorders of the brain”.

Reviewer 3 Report

The review presented by Mills and colleagues aims to strengthen the literature by reviewing the role of the crosstalk and subset control of microglia and myeloid cells in neurological disorders. There is not a review that addresses this issue, what becomes this review an essential tool for this scientific field.

Major concerns:

1.     The reference list is incomplete. There are only 92 references in the list, while during the text, there are at least 167 cited references.

2.     The summary is focused on the last part of the review. The summary should be expanded to include all topics addressed in this review.

3.     The term “resting microglia” is out of use. Please change for homeostatic microglia.

4.     Please, do not use the term “fully activated microglia”. It is a mistake as microglia can acquire many phenotypes in response to different stimuli.

5.     The following period is really unnecessary for this review. The authors should remove it from the introduction. 

Fully activated microglia are of an ameboid morphology, possessing high motility, and phagocytic activity similar to blood-derived macrophages [20-22]. This activated state occurs subsequent to brain infection, due to the release of pathogen-associated molecular patterns (PAMPs), bacterial lipopolysaccharide (LPS), or virus particles, or sterile brain injury, due to the release of damage-associated molecular patterns (DAMPs) or protein aggregates such as amyloid β [23].”

6.     Be careful with the following sentence as it does not always happen in this way. In EAE, for instance, some papers described the release of proinflammatory cytokines from astrocytes are the trigger for peripheral-derived myeloid cells recruitment.

“The inflammatory response initiated by microglia is responsible for the recruitment of peripheral-derived myeloid cells including monocytes [16,46].”

7.     In line 92, please change the term microbes to microorganisms.

8.     Do not describe the M1 and M2 phenotypes in the introduction. It is out of use, simplistic and wrong. Please remove the following text from the introduction.

“However, following insult, it is well-established that microglia diversify their phenotypic identity and function, resulting in an M1-like (pro-inflammatory) or an M2-like (anti-inflammatory) state in response to environmental stimuli [18,25,34,44,81]. M1-like microglia release pro-inflammatory cytokines, such as TNF, IL1β, IL12 and produce reactive oxygen and nitrogen species (ROS and RNS, respectively) that can mediate the recruitment of innate immune cells and induce neurotoxic effects [14]. Activated M1-like microglia express nitric oxide synthase (iNOS), which contributes to neuronal cell death [29]. The M2-like phenotype is more functionally diverse including subsets (i.e. - M2a, M2b, or M2c) that are driven by various cytokines [33,44,82].”

“Traditional classifications in disease consider the general immunological state of microglia as either classical - M1-like - or alternative - M2-like - and have been extensively reviewed elsewhere [33,45,48]. However, this over simplistic classification lumps unique subgroups with distinct functional and immunological signatures.”

9.     Please, identify in table 1 which papers are from human and murine microglia.

10.  Is reference 15 wrongly cited in line 225? Please check.

11.  The section “Peripheral-derived monocytes replace microglia and BAMs” should be revised. There is a lot of information in this section that is not supported by references. Please add references to support your claims.

12.  Still, in regard to this section, the title is misleading. The term “replacement” is misused as the authors affirm, and this referee agrees, that the engrafted peripheral-derived macrophages are different from the native microglia. The title could be replaced by “peripheral-derived macrophages engrafted on diseased brains establish important crosstalks with native microglia”. Or another title, from your preference, that reflects this idea.

13.  The authors should add another example, apart from EAE in section D.

14.  Could be very interesting and useful for this review if the authors add a new section in which authors describe how peripheral-derived macrophages invade the brain parenchyma, taking into consideration the dependence or not of the BBB, and how it occurs when the BBB is intact.

Author Response

Comment 1: The reference list is incomplete. There are only 92 references in the list, while during the text, there are at least 167 cited references.

            Response: Same as reviewer 1, comment 1: We appreciate this comment. It appears that our EndNote tool improperly synced when importing citations, resulting in this discrepancy in citations. We have re-synced our paper with EndNote and the reference list now has 179 references matching both text and references.

Comment 2: The summary is focused on the last part of the review. The summary should be expanded to include all topics addressed in this review.

            Response: We thank the reviewer for this insightful comment, we have additional text in the summary which includes all topics reviewed.

Line 411: “Importantly, future studies may address sex differences and how PDM subsets interact with ontologically distinct microglia to regulate their phenotypic state and whether this cross-talk can be recalibrated to control expression of pro- and anti-inflammatory responses. Overall, resident microglia are key players in the progression and resolution of injury to the CNS.  Microglia production of inflammatory cues aid in the recruitment of peripheral-derived immune cells, including monocyte/macrophages, whose coordination may be central to regulating the neuroinflammatory milieu across numerous neurological disorders. The use of single-cell RNA sequencing has helped detail the complex transcriptomic characteristics of newly identified myeloid derivatives, as well as enhanced our understanding of their origins and local responsibilities across the lifespan.”.

Comment 3: The term “resting microglia” is out of use. Please change for homeostatic microglia.

            Response: We appreciate the reviewer suggesting the need to make this correction and have updated lines 53 and 126 to say “homeostatic microglia” and not “resting microglia”.

Comment 4: Please, do not use the term “fully activated microglia”. It is a mistake as microglia can acquire many phenotypes in response to different stimuli.

            Response: We thank the reviewer for this comment, we agree the nomenclature is inaccurate and have removed mention of this term.

Comment 5: The following period is really unnecessary for this review. The authors should remove it from the introduction.

“Fully activated microglia are of an ameboid morphology, possessing high motility, and phagocytic activity similar to blood-derived macrophages [20-22]. This activated state occurs subsequent to brain infection, due to the release of pathogen-associated molecular patterns (PAMPs), bacterial lipopolysaccharide (LPS), or virus particles, or sterile brain injury, due to the release of damage-associated molecular patterns (DAMPs) or protein aggregates such as amyloid β [23].”

            Response: Thank you for this comment. We have removed this section and updated our citations to reflect this change.

Comment 6: Be careful with the following sentence as it does not always happen in this way. In EAE, for instance, some papers described the release of proinflammatory cytokines from astrocytes are the trigger for peripheral-derived myeloid cells recruitment.

“The inflammatory response initiated by microglia is responsible for the recruitment of peripheral-derived myeloid cells including monocytes [16,46].”

            Response: Thank you very much for bring this to our attention.  We changed the text as follows: “The inflammatory response initiated by CNS glia is responsible for the recruitment of peripheral-derived myeloid cells including monocytes”

Comment 7: In line 92, please change the term microbes to microorganisms.

            Response: Thank you for your comment. We have updated line 92 (now line 103 following corrections) to read “passage of microorganisms” not “passage of microbes”.

Comment 8: Do not describe the M1 and M2 phenotypes in the introduction. It is out of use, simplistic and wrong. Please remove the following text from the introduction.

“However, following insult, it is well-established that microglia diversify their phenotypic identity and function, resulting in an M1-like (pro-inflammatory) or an M2-like (anti-inflammatory) state in response to environmental stimuli [18,25,34,44,81]. M1-like microglia release pro-inflammatory cytokines, such as TNF, IL1β, IL12 and produce reactive oxygen and nitrogen species (ROS and RNS, respectively) that can mediate the recruitment of innate immune cells and induce neurotoxic effects [14]. Activated M1-like microglia express nitric oxide synthase (iNOS), which contributes to neuronal cell death [29]. The M2-like phenotype is more functionally diverse including subsets (i.e. - M2a, M2b, or M2c) that are driven by various cytokines [33,44,82].”

“Traditional classifications in disease consider the general immunological state of microglia as either classical - M1-like - or alternative - M2-like - and have been extensively reviewed elsewhere [33,45,48]. However, this over simplistic classification lumps unique subgroups with distinct functional and immunological signatures.”

            Response: This text has been updated on line 138-149.

Comment 9: Please, identify in table 1 which papers are from human and murine microglia.

            Response: The table title highlights that only human tissue is reviewed “Human gene signatures…” We also reiterate this in the column 2 heading.

Comment 10: Is reference 15 wrongly cited in line 225? Please check.

            Response: Thank you for pointing this error out, upon update of the citations, we believe that this matter has been resolved and that the appropriate references are included.

Comment 11: The section “Peripheral-derived monocytes replace microglia and BAMs” should be revised. There is a lot of information in this section that is not supported by references. Please add references to support your claims.

            Response: Thank you for pointing this matter out, we have resolved this issue by properly citing and including references for all statements. 

Comment 12: Still, in regard to this section, the title is misleading. The term “replacement” is misused as the authors affirm, and this referee agrees, that the engrafted peripheral-derived macrophages are different from the native microglia. The title could be replaced by “peripheral-derived macrophages engrafted on diseased brains establish important crosstalks with native microglia”. Or another title, from your preference, that reflects this idea.

            Response: We agree and have updated the title as such: “External factors influencing CNS engraftment of peripheral-derived monocytes

Comment 13: The authors should add another example, apart from EAE in section D.

            Response: We thank the reviewer for making this suggestion. We included additional studies in stroke comparing the multiple myeloid lineage subsets using scRNAseq to line 337-140.

Comment 14: Could be very interesting and useful for this review if the authors add a new section in which authors describe how peripheral-derived macrophages invade the brain parenchyma, taking into consideration the dependence or not of the BBB, and how it occurs when the BBB is intact.

            Response: We agree and have added the following paragraph on transport across an intact BBB to section 1c (lines 275-284): “In non-pathologic states, the brain parenchyma is a highly immune-privileged tissue separated from the blood via the BBB, a non-fenestrated endothelial barrier with tight junctions composed of transmembrane proteins such as claudins, occludins and junctional adhesion molecules. Yet in the absence of neuroinflammation, perivascular BAMs and microglia interact with peripheral-derived T cells via antigen presentation to facilitate immune surveillance. Activated T cells then cross the BBB endothelium via paracellular (tight junction-dependent) or transcellular (endothelial cell receptor and charge-dependent) transport mechanisms, while preserving the integrity of the BBB. As neuroinflammation progresses, chemoattractants (e.g., chemokine-like factor-1) mediate neutrophil migration, receptors (e.g., CCR2) recruit monocytes, and endothelial cell receptor expression (e.g., P-selectin, ICAM-1) promotes paracellular permeability, causing blood-borne immune cell infiltration into a disrupted BBB.”

Round 2

Reviewer 2 Report

The authors have addressed my concerns.